# Implementation of Machine Learning Models to Ensure Radiotherapy Quality for Multicenter Clinical Trials: Report from a Phase III Lung Cancer Study

**DOI:** 10.3390/cancers15041014

**Published:** 2023-02-05

**Authors:** Huaizhi Geng, Zhongxing Liao, Quynh-Nhu Nguyen, Abigail T. Berman, Clifford Robinson, Abraham Wu, Romaine Charles Nichols Jr, Henning Willers, Nasiruddin Mohammed, Pranshu Mohindra, Ying Xiao

**Affiliations:** 1Department of Radiation Oncology, University of Pennsylvania, Philadelphia, PA 19104, USA; 2The University of Texas MD Anderson Cancer Center, Houston, TX 77054, USA; 3Siteman Cancer Center, Washington University, Saint Louis, MO 63110, USA; 4Memorial Sloan-Kettering Cancer Center LAPS, New York, NY 10065, USA; 5Department of Radiation Oncology, University of Florida Health Science Center-Gainesville, Jacksonville, FL 32610, USA; 6Dana-Farber/Partners Cancer Care LAPS, Boston, MA 02215, USA; 7Northwestern Medicine Cancer Center Warrenville, Warrenville, IL 60555, USA; 8Greenebaum Cancer Center, University of Maryland, Baltimore, MD 21201, USA

**Keywords:** radiotherapy, knowledge-based planning, non-small-cell lung cancer, clinical trial quality assurance

## Abstract

**Simple Summary:**

Over 50% of all cancer patients receive radiation therapy (RT). The quality of the RT treatment plan is directly related to patient outcomes, such as overall survival and complications related to RT. In this study, we explore a knowledge-based machine learning tool for RT plan quality evaluation on plans submitted to a multicenter non-small-cell lung cancer clinical trial. The results of this study may provide critical information for the analysis of the end points of the trial. This study also demonstrated the feasibility of using this novel tool for RT plan quality assessment in the multicenter clinical trial setting.

**Abstract:**

The outcome of the patient and the success of clinical trials involving RT is dependent on the quality assurance of the RT plans. Knowledge-based Planning (KBP) models using data from a library of high-quality plans have been utilized in radiotherapy to guide treatment. In this study, we report on the use of these machine learning tools to guide the quality assurance of multicenter clinical trial plans. The data from 130 patients submitted to RTOG1308 were included in this study. Fifty patient cases were used to train separate photon and proton models on a commercially available platform based on principal component analysis. Models evaluated 80 patient cases. Statistical comparisons were made between the KBP plans and the original plans submitted for quality evaluation. Both photon and proton KBP plans demonstrate a statistically significant improvement of quality in terms of organ-at-risk (OAR) sparing. Proton KBP plans, a relatively emerging technique, show more improvements compared with photon plans. The KBP proton model is a useful tool for creating proton plans that adhere to protocol requirements. The KBP tool was also shown to be a useful tool for evaluating the quality of RT plans in the multicenter clinical trial setting.

## 1. Introduction

About 50% of all cancer patients receive radiation therapy (RT). RT treatment plans that do not follow specific guidelines are associated with a lower survival probability [1,2], higher probability of disease progression [3] or increased risk of RT-related complications [4,5,6]. Therefore, quality assurance (QA) of RT plans is critical for patient care. Additionally, it is also essential for the success of clinical trials with an RT component. 

Accurate delineation of the plan target volumes and adjacent organs at risk (OARs) is the prerequisite of a high-quality plan, which is beyond the scope of this study. Assuming the accuracy of all the delineations, the proper optimization of a plan by delivering uniform prescription doses to the target while mitigating dose deposition on the critical OARs is the main goal and therefore quality evaluation criterion. The treatment plan optimization process is a complicated process involving the skills and experiences of the planners. The Imaging Radiation Oncology Core (IROC) of the National Clinical Trials Network performed a QA review of the plans. All cases were subjected to the population-based dose constraint criteria defined in the trial protocol, which were based on past experiences of clinicians. Although this process identifies the plan that violates protocol constraints, it does not indicate the underlying reasons for the violation or the potential for quality improvement. A Knowledge-Based Planning (KBP) model uses a library of high-quality plans to provide a set of mathematical models between individual anatomy and the lowest achievable dose profiles to the OARs [7,8]. Personalized optimal treatment plans can be realized with the model predictions as plan optimization guidance. The model-generated predictions of dose profiles, when compared with submitted plan doses, can also serve as peer review of the plan quality. KBP for photon therapy is widely adopted in clinical settings [9,10,11,12,13,14], however, it is still in a novel stage for the RT QA in multicenter clinical trial settings. Several reports have shown the potential of this technique for the clinical trial QA of intensity-modulated radiation therapy (IMRT) [3,15,16,17,18,19,20,21]. 

The installation of proton accelerators has increased substantially in recent years [22]. The physics of proton beam promises overall lower OAR doses. There is some clinical evidence of the dosimetric advantages of proton therapy vs. conventional photon therapy [23,24,25]. However, large-scale randomized clinical trials are needed to prove these advantages. With the complication of proton planning, proton KBP as an emerging technique is also under investigations [26,27,28,29,30]. No study has been reported for the use of proton KBP for clinal trial QA purposes. 

The Radiation Therapy Oncology Group (RTOG) 1308 is a randomized phase III trial that compares overall survival after photon versus proton for inoperable stage II–III non-small-cell lung cancer (NSCLC) receiving concurrent chemotherapy and RT. The main goal of the trial is to see if proton therapy can improve overall survival compared with IMRT by lowering the risk of severe OAR toxicity. QA of the treatment plans submitted to both the photon and proton arms is essential for a fair comparison of these two modalities.

In this study, we used the KBP method for the QA of RTOG 1308 plans, with a focus on reporting the general quality of the plans submitted to both the photon and proton arms. Moreover, this is the first experience utilizing the proton KBP model for the QA of multicenter clinical trial proton plans. 

## 2. Materials and Methods

The process of data selection, model training, and application of planned QA are described in this section.

### 2.1. Initial Data Review and Selection

Data from 210 patients enrolled in RTOG1308 at the time of this study were evaluated according to the IROC QA procedure. The treatment arm (photon or proton), the technique (passive scattering (PS) or intensity-modulated proton therapy (IMPT)), the type of treatment machine, and the dosimetric review in accordance with protocol dose constraints (*per protocol: score 1, variation acceptable: score 2, and deviation unacceptable: score 3*) were all collected. Table 1 summarizes the protocol’s dosimetric constraints for performing the initial plan quality review. The review revealed that there were no deviation unacceptable cases and five score 2 cases among all IMPT cases; 5 deviation unacceptable cases and 9 variation acceptable cases among all PS cases; and 4 deviation unacceptable cases and 6 variation acceptable cases among all photon cases. 

Following the initial assessment, 130 patient data sets were chosen for this investigation. Fifty score 1 photon cases were randomly selected for model training. Eighty patients were selected as testing cases, and all cases of score 2 and score 3 were included with preferences. Among the 80 testing cases, 20 received IMPT, 20 received PS, and 40 received photon treatments. DICOM CT and RT structures of these 130 patients were imported into Eclipse Treatment Planning System (TPS) (Varian Medical Systems, Palo Alto, CA, USA).

### 2.2. Model Training

Original score 1 photon plans on the 50 training cases were used for initial photon model training (DVH estimation algorithm version 15.7.02). 

IMPT plans to treat NSCLC with 2Gy/fraction, 35 fractions were generated manually in these 50 patients with ProBeam beam data. The dose distribution was optimized using fluence-base nonlinear universal proton optimizer (NUPO 16.0.2). The spot spacing was 0.425 of the energy-dependent in-air full-width half-maximum spot size at the isocenter. The multifield simultaneous spot optimization method was selected for all plans. A 5 cm range shifter was used for all fields. The Proton Convolution Superposition algorithm was used for the final dose calculation with a grid of 2.5 × 2.5 × 2.5 mm^3^. A constant relative biological effectiveness (RBE) of 1.1 was applied.

The manually generated 50 IMPT plans were reviewed as per protocol and were included for the initial proton model training (DVH estimation algorithm 16.0.2). 

Figure 1 illustrates the model training workflow for the photon and proton arms. Initial models were trained with reviewed per protocol plans. Plan optimization iterations were carried out as indicated in Figure 1 to ensure the optimal quality of the model plans. After two optimization iterations, the final models were generated. 

The original plans submitted by multiple institutions used a variety of treatment planning systems (Pinnacle3, Elicpse, Raystation, and Elekta XIO) and were based on a variety of treatment machines (Photon: Varian LINACs (Clinac, Trubeam), Elecka. Proton: IBA, Hitachi, Mevion, ProBeam). The original plan beam angles were obtained from the DICOM headers of the submitted RT plans. A plan was created that used the original beam setup with the original RT dose file imported and attached to the plan. The photon plans were duplicated and then reoptimized based on the KBP models using the 10 MV beam models Clinac 23EX 15.6.03 ABX, with Millennium_120 leaf. Additionally, the same settings described in the model training section were used for the proton replan with the original submitted plan beam angles and model-based optimizations. 

Model optimization priorities were manually adjusted on selected testing cases (with challenging anatomy and large tumor volume) for several testing runs.

### 2.3. Plan Quality Review by Models

The submitted plans were compared with the KBP plans dosimetrically. The general quality differences between the originally submitted plans and the KBP plan were analyzed using mean dosimetric points and the *t*-test. Individual score 3 plans were also examined to determine whether there was a possibility for quality improvement.

## 3. Results

### 3.1. Final Model Settings

The KBP platform provides model settings for treatment planning optimization objectives and priorities of the objectives. These model objectives were fine-tuned to produce a plan with uniform target dose coverage and optimized OAR sparing after a single iteration of optimization. The finalized priority settings are reported in Table 2. Both models will be published for researchers and clinicians to access for the optimization of the RT plan and QA purposes.

### 3.2. Proton KBP Model Evaluation

All initial manual IMPT plans built on the 50 training patients met all protocol dose constraints (Table 1). We performed two iterations of plan optimizations, followed by addition of the reoptimized plans for training new models to remove potential dosimetric outliers and enhance the model performance. The results of the dosimetric comparison and the *t*-test of 50 manual IMPT and the final KBP plans are shown in Table 3. Compared with manual plans, KBP plans demonstrate statistically significant improvements in OAR protection while maintaining the same or greater target coverage, demonstrating the efficacy of the KBP tool for IMPT plan optimizations. 

### 3.3. Plan Quality Review

We used the models to reoptimized 40 photon plans and 40 proton plans submitted to RTOG 1308. Additionally, we compared the model-based plans with the original submitted plans to evaluate the quality of the plans from both photon and proton arms.

#### 3.3.1. Photon Plan Quality Review

The dosimetric comparison between the KBP photon plans and the submitted photon plans, together with the results of the *t*-test, are reported in Table 4. The KBP photon plans demonstrate overall optimized heart and lung doses without significant changes at other dosimetric points. There were six-variation (score 2) and four-deviation (score 3) photon plans among the forty testing plans. Two out of the six score 2 plans were improved to score 1 plans. Only one score 3 plan was improved to score 2. Three out of the four score 3 plans were analyzed to be of good quality; no further optimization was realized.

#### 3.3.2. Proton Plan Quality Review

The critical dosimetric points comparison between KBP IMPT and the original proton plans submitted was listed separately for the original IMPT and PS cases in Table 5 due to the intrinsic difference of PS plans and IMPT plans. KBP IMPT plans demonstrate statistically significant improvements in target coverage (PTV D99%[Gy] indicates target dose coverage) and OAR sparing (especially lower heart and lung doses) for both cohorts of patients. All KBP IMPT plans in the 40 testing patients met all protocol dose constraint criteria given in Table 1, including the original five variation acceptable IMPT plans, nine variation acceptable, and five deviation unacceptable PS plans submitted. 

Box plots were also generated to show the dosimetric points comparison between the original plans and the KBP IMPT plans for the two cohorts of patients in Figure 2. Box plots show that the original submitted proton plans (both IMPT and PS) vary in target coverage (PTV D99%[Gy]) with several PS plans deviating from the average PTV D99%[Gy] by up to 10 to 15 Gy. KBP IMPT plans provide more uniform target coverage while reducing overall doses to the heart, lungs, and esophagus. Figure 3 shows the screen capture of the 3D dose wash comparison of the original IMPT plan versus the KBP IMPT plan. The KBP IMPT plan significantly reduced dose spillage to normal lung tissue and improved dose uniformity to the tumor.

## 4. Discussions

We attempted a comparison between plans generated with photon vs. proton beams for some of the cases. KBP IMPT plans were created in 40 testing patients from the photon cohort, with detailed findings reported in the Appendix A section. Generally, the findings demonstrate the dosimetric superiority of proton therapy compared with photon therapy. However, to realize this dosimetric superiority, optimal proton plan quality is required. Although the plans submitted to both arms are of acceptable quality, proton plans exhibit a greater degree of variation in quality and indicate greater room for improvement. The results of this study may provide critical information for the analysis of the trial end point. 

Due to the limitations of the double scatter techniques, some initial PS plans failed to meet the protocol dose constraints. KBP IMPT, on the other hand, easily met the protocol criteria for those patients. This could imply that IMPT has inherent dosimetric advantages over PS in certain patients with difficult anatomy. All proton plans submitted to this trial five years ago utilized the PS method; however, the majority of recent submissions utilized IMPT. This demonstrates the evolution of proton treatment techniques.

IMRT plans may not be subject to significant quality variations caused by treatment machines. However, the quality of the IMPT plan is influenced by the treatment machine’s beam quality, spot size, range modulators, and original beam energy ranges. The model used in this investigation was trained using plans constructed from beam models provided by the treatment planning system manufacturer as golden beam models. Due to the limitations of the treatment machines, the plan quality achieved in this study may not be replicable in the enrolling centers. The test results presented in this study just indicate the feasible plan quality with the beam models and techniques; they do not indicate the specific causes of the variation in plan quality.

## 5. Conclusions

This study summarizes the general quality of the RT plans submitted to multicenter clinical trials. KBP models were used to conduct a more thorough review of the quality of the plan and the potential for improvement. Proton plans, a relatively emerging technique, show more variation in quality than photon plans, which are consistent with good quality and have little room for improvement with existing approaches. The KBP IMPT model is a useful tool to create IMPT plans that adhere to protocol requirements. The KBP tool was also shown to be a helpful tool for reviewing the quality of RT plans in the multicenter clinical trial setting. Both photon and proton models built using multicenter clinical trial data in this study will be published to researchers as well as clinicians to access for RT plan optimization and QA purposes.

## Figures and Tables

**Figure 1 cancers-15-01014-f001:**
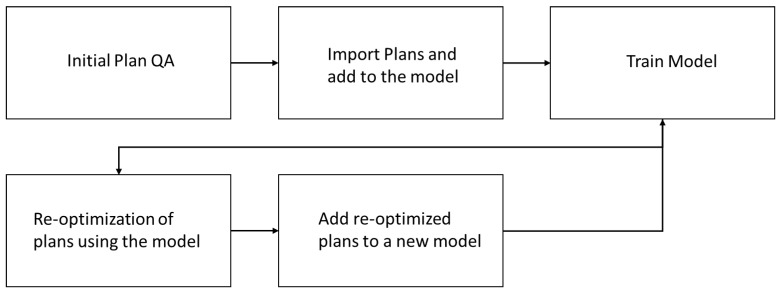
Workflow chart for model training.

**Figure 2 cancers-15-01014-f002:**
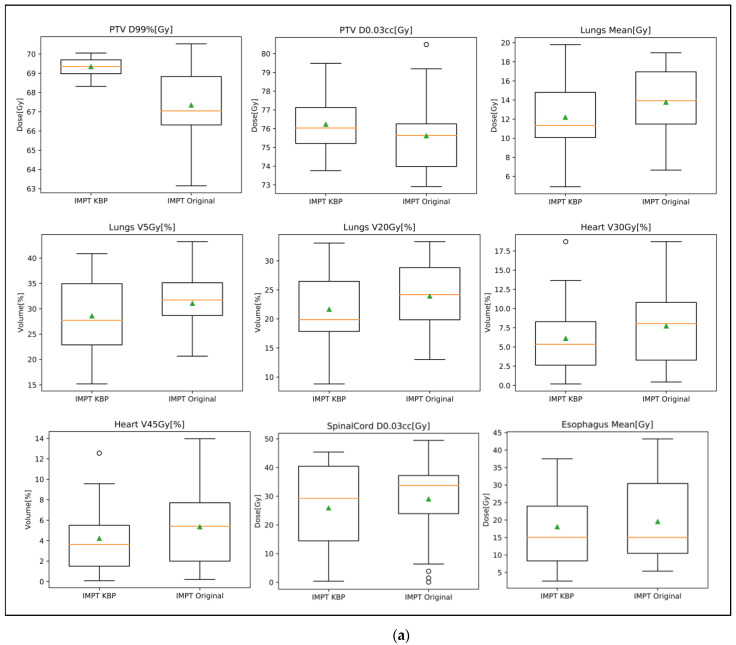
Box plots of dosimetric comparison for IMPT KBP plans versus original proton plans: (**a**) IMPT KBP plans versus original IMPT plans; (**b**) IMPT KBP plans versus original PS plans. (Orange lines indicate the average values and the green triangles indicate the median values; Circles indicate potential outliers).

**Figure 3 cancers-15-01014-f003:**
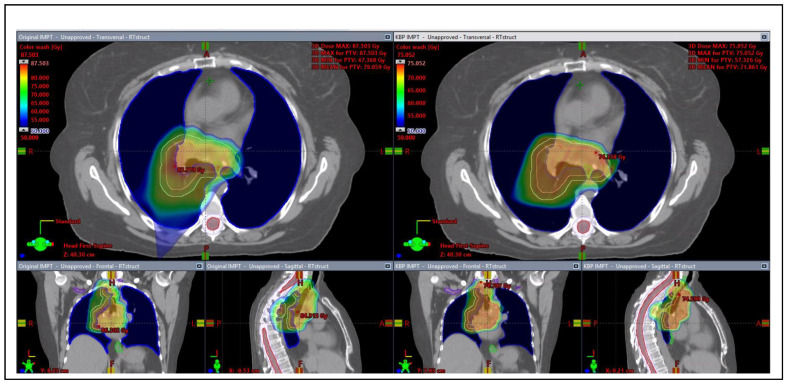
Screen capture of dose color wash comparison of a sample KBP IMPT plan (**right**) versus original IMPT plan (**left**).

**Table 1 cancers-15-01014-t001:** QA dosimetric criteria for the RTOG 1308 plan.

Structures	Dosimetric Points	Per Protocol Value	Acceptable Variation Value
PTV	D95%[Gy]	≥Rx	≥95% Rx
	D0.03cc[Gy]	<=110% Rx	<=120% Rx
	D99%[Gy]	>=80% Rx	>=80% Rx
Heart	V45Gy[%]	<=35	<=40
	V30Gy[%]	<=50	<=55
Lungs	Mean[Gy]	<=20	<=22
	V20Gy[%]	<=37	<=40
	V5Gy[%]	<=60	<=65
Esophagus	V74Gy[cc]	<=1	<=1.5
Spinal Cord	D0.03cc[Gy]	<=50	<=52

**Table 2 cancers-15-01014-t002:** Model Optimization and Priority Settings.

Structures	Objectives	Priorities
PTV	Dmin ≥ 103% Rx dose	150
Esophagus	Dmax ≤ 110% Rx dose	100
	Dmax ≤ 70 Gy	80
	Line (preferring target)	Generated priority
Brachial Plexus	Dmax ≤ 60 Gy	80
Heart	V30Gy ≤ Generated Value	Generated priority
	Line (preferring target)	Generated priority
Lungs	Dmax ≤ 65 Gy	100
	Line (preferring target)	Generated priority
Spinal cord	Dmax ≤ 45 G	100
	Line (preferring target)	Generated priority

**Table 3 cancers-15-01014-t003:** Dose comparison and *t*-test results of 50 manual IMPT plans and 50 KBP plans in the same patients.

Structure Name	Dose Point	Manual	KBP	*p* Value
PTV	D95%[Gy]	69.43 ± 0.52	70.20 ± 0.29	**<0.0001**
	D0.03cc[Gy]	79.65 ± 1.37	79.73 ± 1.63	0.4345
	D99%[Gy]	67.74 ± 0.93	68.89 ± 0.74	**<0.0001**
Heart	V45Gy[%]	6.2 ± 6.1%	4.9 ± 5.0%	**<0.0001**
	V30Gy[%]	8.6 ± 7.5%	7.0 ± 6.4%	**<0.0001**
	Mean[Gy]	6.35 ± 5.07	5.26 ± 4.34	**<0.0001**
Lungs	V5Gy[%]	31.8 ± 8.7%	31.0 ± 8.9%	**0.0022**
	V20Gy[%]	23.0 ± 6.1%	22.5 ± 6.4%	**0.0039**
	Mean[Gy]	13.11 ± 3.38	12.81 ± 3.64	**0.0028**
Esophagus	V74Gy[cc]	0 ± 0	0 ± 0	N/A
Spinal Cord	D0.03cc[Gy]	41.06 ± 9.80	40.58 ± 10.63	0.3579

**Table 4 cancers-15-01014-t004:** Dose comparison and *t*-test results of forty submitted IMRT versus KBP IMRT plans. (All IMRT plans were normalized so that 70Gy will cover 95% of the PTV volume.)

Structure Name	Dose Point	Photon Original	IMRT KBP	*p* Value
PTV	D95%[Gy]	70	70	N/A
	D0.03cc[Gy]	78.80 ± 3.06	81.52 ± 2.31	0.447
	D99%[Gy]	67.86 ± 0.98	66.14 ± 1.47	0.7081
Heart	V45Gy[%]	10.1% ± 9.2%	13.1 ± 9.8%	**0.0074**
	V30Gy[%]	16.5% ± 12.9%	21.9 ± 17.0	0.1029
	Mean[Gy]	14.02 ± 8.66	13.71 ± 12.31	0.0901
Lungs	V5Gy[%]	52.3% ± 10.1%	53.4 ± 13.1%	0.1007
	V20Gy[%]	28.9% ± 6.2%	30.2 ±7.7%	0.8153
	Mean[Gy]	17.85 ± 3.69	17.23 ± 3.39	**0.0057**
Esophagus	V74Gy[cc]	0.20 ± 0.45	0 ± 0	N/A
Spinal cord	D0.03cc[Gy]	43.97 ± 7.04	42.44 ± 4.51	0.5377

**Table 5 cancers-15-01014-t005:** Dose comparison and *t*-test results of twenty IMPT plans submitted and twenty PS plans versus KBP IMPT plans submitted in the same patients.

Structure	Dose Point	IMPT Original	IMPT KBP	*p* Value	PS Original	IMPT KBP	*p* Value
PTV	D95%[Gy]	69.60 ± 1.26	70.37 ± 0.16	**0.0127**	69.44 ± 3.39	70.30 ± 0.44	0.2497
	D0.03cc[Gy]	75.62 ± 2.09	76.23 ± 1.54	0.2999	79.249 ± 3.97	76.11 ± 1.29	**0.0053**
	D99%[Gy]	67.34 ± 2.10	69.34 ± 0.48	**0.0006**	65.50 ± 5.08	69.14 ± 0.73	**0.0044**
Heart	V45Gy[%]	5.4% ± 3.9%	4.2% ± 3.3%	**0.0043**	5.6% ± 5.5%	4.2% ± 4.4%	**0.0001**
	V30Gy[%]	7.7% ± 5.2%	6.1% ± 4.6%	**0.0048**	7.6% ± 6.7%	5.9% ± 5.6%	**0.0003**
	Mean[Gy]	6.25 ± 4.07	4.92 ± 3.54	**0.0032**	5.46 ± 4.74	4.60 ± 4.12	**0.0208**
Lungs	V5Gy[%]	31.1% ± 6.2%	28.6% ± 7.6%	**0.0111**	33.4% ± 10.8%	28.8% ± 9.4%	**<0.0001**
	V20Gy[%]	23.9% ± 6.0%	21.6% ± 6.6%	**0.0063**	25.9% ± 7.9%	22.04% ± 8.14%	**<0.0001**
	Mean[Gy]	13.76 ± 3.61	12.19 ± 3.98	**0.002**	14.64 ± 4.43	12.68 ± 4.43	**<0.0001**
Esophagus	V74Gy[cc]	0.05 ± 0.22	0.01 ± 0.04	N/A	0.80 ± 2.48	0.01 ± 0.03	N/A
Spinal Cord	D0.03cc[Gy]	29.02 ± 14.80	25.90 ± 15.52	0.2197	27.26 ± 16.55	33.20 ± 13.75	**0.0466**

## Data Availability

3rd Party Data: Restrictions apply to the availability of these data. Data were obtained from Imaging Oncology Core Radiotherapy Quality Assurance team. This is an ongoing trial; no data will be made available to the public before any publication of the end point of this trial. After the closure and publication of the endpoint of this trial, data can be applied via data sharing through NRG oncology.

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
