# Peer review of "Implementation of Machine Learning Models to Ensure Radiotherapy Quality for Multicenter Clinical Trials: Report from a Phase III Lung Cancer Study"

_cancers, 2023, doi:10.3390/cancers15041014_

Round 1

Reviewer 1 Report

This study evaluated the usefulness of KBP in QA for RT planning for clinical trials using the prospective study cohort. Moreover, authors showed KBP may be helpful to improve proton beam therapy plan by KBP IMPT plan. This study may have readers’ interest of Cancers.

I just have following minor concerns.

Minor

1.     In abstract 25th line, “Knowledge-based machine learning (KBP) models” may need to be revised.

2.     In Model Training 100th line, 2.5×2.5×2.5mm3  2.5×2.5×2.5mm3

3.     In results section, “3.1 Final Model Settings” is hard to understand. Please revise this paragraph.

4.     In results 139th line, “Iteration”  iteration

Author Response

Thank you for the review. I have responded all the minor concerns raised by you in the revision.

Reviewer 2 Report

Main problem of this paper is more ethca ethical point. The first paper RTOG 1308 has not been published. Therefore this paper should not be published without NRG approval.

in the introduction, there is no reference of RTOG 1308.

Author Response

Thank you for the comments. The manuscript is actually approved by NRG publications and was assigned as M17354. Although the study used data from an ongoing trial without primary endpoint publication, the results from this study addresses a sencondary QA endpoint and was reviewed by NRG publications and NCI committee. Attached is the author line approval form from NRG publications.

The introduction part was re-written with more references added. The whole manuscript was also checked for language issues.

Reviewer 3 Report

This is an interesting paper concerning the high promising concept of machine learning in radiotherapy.

Although the paper is well written, it is very difficult to read and to understand as a clinician. Please describe more extensive and in basic non technic language the aim and results

Author Response

Thank you for the review. The introduction part was re-written by adding some back ground knowledge for better understanding the aim of this study. More references was also added. The results part was not rearranged since those were simple dose comparisons of KBP based plans with manual plans and submitted plans for different cohort of patiens. Please let me know if any part of the results is confusing to you. Minor language issues were address also.

Round 2

Reviewer 2 Report

This manuscript has been revised and substantially improved.

It will be acceptable for publication .

Author Response

Thank you for the review! The data used are collected from multicenter clinical trial network and are subjected to centra IRB approval. Please see attached trial protocol (p32 for CIRB information) and blank patient consent form. 
